# *APOBEC3B* Gene Expression in Ductal Carcinoma In Situ and Synchronous Invasive Breast Cancer

**DOI:** 10.3390/cancers11081062

**Published:** 2019-07-27

**Authors:** Anieta M. Sieuwerts, Shusma C. Doebar, Vanja de Weerd, Esther I. Verhoef, Corine M. Beauford, Marie C. Agahozo, John W.M. Martens, Carolien H.M. van Deurzen

**Affiliations:** 1Department of Medical Oncology and Erasmus MC Cancer Institute, 3015 GD Rotterdam, The Netherlands; 2Cancer Genomics Netherlands, Erasmus MC Cancer Institute, 3015 GD Rotterdam, The Netherlands; 3Department of Pathology, Erasmus MC Cancer Institute, 3015 GD Rotterdam, The Netherlands

**Keywords:** APOBEC3B, gene expression, breast cancer, ductal carcinoma in situ, infiltrating breast cancer, PIK3CA

## Abstract

The underlying mechanism of the progression of ductal carcinoma in situ (DCIS), a non-obligate precursor of invasive breast cancer (IBC), has yet to be elucidated. In IBC, Apolipoprotein B mRNA Editing Enzyme, Catalytic Polypeptide-Like 3B (APOBEC3B) is upregulated in a substantial proportion of cases and is associated with higher mutational load and poor prognosis. However, APOBEC3B expression has never been studied in DCIS. We performed mRNA expression analysis of *APOBEC3B* in synchronous DCIS and IBC and surrounding normal cells. RNA was obtained from 53 patients. The tumors were categorized based on estrogen receptor (ER), progesterone receptor (PR), human epidermal growth factor receptor 2 (Her2) and phosphoinositide-3-kinase, catalytic, alpha polypeptide (PIK3CA) mutation status. *APOBEC3B* mRNA levels were measured by RT-qPCR. The expression levels of paired DCIS and adjacent IBC were compared, including subgroup analyses. The normal cells expressed the lowest levels of *APOBEC3B*. No differences in expression were found between DCIS and IBC. Subgroup analysis showed that *APOBEC3B* was the highest in the ER subgroups of DCIS and IBC. While there was no difference in *APOBEC3B* between wild-type versus mutated PIK3CA DCIS, *APOBEC3B* was higher in wild-type versus PIK3CA-mutated IBC. In summary, our data show that *APOBEC3B* is already upregulated in DCIS. This suggests that APOBEC3B could already play a role in early carcinogenesis. Since APOBEC3B is a gain-of-function mutagenic enzyme, patients could benefit from the therapeutic targeting of APOBEC3B in the early non-invasive stage of breast cancer.

## 1. Introduction

Ductal carcinoma in situ (DCIS) is a non-obligate precursor of invasive breast cancer (IBC) [1]. This is supported by previous studies that reported a high genomic concordance of synchronous DCIS and IBC [2,3,4]. However, despite molecular similarities, recent in-depth genetic studies also reported specific mutations that were either restricted to the in situ or the invasive component [3,5]. Increased insight in the molecular changes during DCIS progression has the potential to reveal novel, potentially targetable drivers of progression.

A major role of Apolipoprotein B mRNA Editing Enzyme, Catalytic Polypeptide-Like 3B (APOBEC3B) has been reported in breast cancer and several other cancers [6,7,8,9]. This enzyme is a member of the APOBEC family of deaminases and is involved in DNA cytosine deaminase activity, which has diverse biological functions, including activities in the innate immune system by restricting virus replication [10]. The upregulation of APOBEC3B is correlated with increased C-to-T transitions and increased mutational load, including known driver mutations in PIK3CA and tumor protein 53 (TP53) [10,11,12]. *APOBEC3B* mRNA is upregulated in a substantial proportion of IBC cases and an association with poor clinical outcome has been reported in Estrogen receptor (ER)-positive subtypes [13]. In addition, we recently reported higher mRNA levels of *APOBEC3B* in breast cancer metastasis as compared to the corresponding primary tumor, which implied that breast cancer progression is associated with the upregulation of APOBEC3B [14]. 

In this study we investigated *APOBEC3B* mRNA expression levels in synchronous DCIS and IBC and correlated the expression with PIK3CA mutation status in order to increase our understanding regarding the expression levels of this enzyme during progression from the in situ to the invasive stage. We believe this could improve breast cancer care in the future since APOBEC3B is a gain-of-function mutagenic enzyme, so patients could potentially be treated with small molecules at a very early, non-invasive stage.

## 2. Results

### 2.1. General Clinicopathological Data

In total, 53 patients were included. Table 1 provides an overview of the clinicopathological data of all patients. The overall median age was 53 years (range 28–102 years). The majority of DCIS and IBC samples were high grade (62.3 and 54.7%, respectively). There was no difference in grade between DCIS and adjacent IBC (Fisher Exact Probability Test *p* = 0.92). Based on immunohistochemical staining, IBCs were categorized into the following five breast cancer subtype categories: ER+/PR high/Her2− (*n* = 13), ER+/PR− or low/Her2− (*n* = 12), ER+/any PR/Her2+ (*n* = 11), ER−/PR−/Her2+ (*n* = 8), or ER−/PR−/Her2− (*n* = 9).

### 2.2. APOBEC3B Expression in Synchronous Normal, DCIS and IBC Cells

Both the Kruskal-Wallis Test and the Median Test indicated that there was a significant difference (*p* < 0.001) in APOBEC3B mRNA levels between the normal controls, DCIS and IBC. APOBEC3B mRNA was lower expressed in the normal mammary epithelial tissue adjacent DCIS and IBC (unpaired Mann-Whitney *U* Test and paired Wilcoxon Signed Ranks Test *p* < 0.001) (Figure 1). There was no statistically significant difference in APOBEC3B mRNA expression between DCIS and IBC (unpaired Mann–Whitney *U* Test *p* = 0.065 (Figure 1), Wilcoxon Signed Ranks Test *p* = 0.082). (Figure 2).

### 2.3. APOBEC3B mRNA Subgroup Analysis

Previous studies reported elevated *APOBEC3B* mRNA levels in breast cancers with otherwise aggressive characteristics, including high histological grade and lack of estrogen expression [7,13,15]. For both DCIS and IBC, there was no correlation between APOBEC3B expression levels and tumor diameter (Spearman Rank Correlation Test *p* > 0.05) or histological grade (Kruskal-Wallis Test *p* > 0.05). Our breast cancer subtype analysis showed that the expression of APOBEC3B was the highest in the ER− subgroup (Mann–Whitney *U* Test, *p* = 0.037) (Figure 3).

### 2.4. APOBEC3B Expression in Epithelial Versus Inflammatory Cells

Based on the positive correlation between APOBEC3B and marker for epithelial content (EPCAM) mRNA levels (Spearman Rank Correlation test, *p* = 0.005 for DCIS, *p* = 0.001 for IBC), APOBEC3B mRNA was mostly expressed by epithelial cells. Of note, there was no significant difference in the levels of EPCAM mRNA between DCIS and synchronous IBC (Wilcoxon Signed Ranks Test, *p* = 0.18).

Since inflammatory cells also express APOBEC3B [16], we investigated whether the number of inflammatory cells could have biased our results by comparing Protein Tyrosine Phosphatase Receptor Type C (PTPRC, gene for the common leukocyte antigen CD45) mRNA levels from DCIS and IBC. There was no correlation between APOBEC3B and PTPRC mRNA levels (Spearman Rank Correlation test, *p* = 0.18 for DCIS and *p* = 0.29 for IBC). However, IBC expressed slightly higher levels of PTPRC when compared with DCIS (Wilcoxon Signed Ranks Test, *p* = 0.023). 

### 2.5. APOBEC3B Expression and PIK3CA Mutation Status

In a recently published study [17], we detected a PIK3CA somatic hotspot mutation in 24.7% (18 out of 73) patients. For these 18 PIK3CA-positive patients, a significantly higher PIK3CA variant allele frequency (VAF) was detected in the DCIS component (45.8%) when compared with the synchronous IBC component (31.7%) (*p* = 0.007). For the *n* = 14 PIK3CA mutation-positive patients (26.4%) included in the current study, a significantly higher PIK3CA VAF was also detected in the DCIS component (52.3%) when compared with the synchronous IBC component (37.2%) (*p* = 0.027). The correlation of PIK3CA VAF with APOBEC3B showed a negative Spearman Rank correlation in IBC (rs = −0.33, *p* = 0.001, *n* = 53). For the DCIS cases, there was no such correlation (rs = 0.02, *p* = 0.89, *n* = 53). Analyzing these data irrespective of the degree of the PIK3CA VAF levels revealed that for the 53 patients analyzed in this study, APOBEC3B mRNA levels in IBC were significantly lower in the eight patients with exon 9 (G to A)-mutated PIK3CA when compared with the *n* = 39 wild-type PIK3CA cases (Mann-Whitney *U* test *p* = 0.017). No such difference was observed for the DCIS cases (*p* = 0.28) (Figure 4). Albeit not statistically significant, APOBEC3B mRNA levels were higher overall in the *n* = 39 PIK3CA wild-type IBC samples when compared with the PIK3CA wild-type DCIS samples (Mean ± SEM: −4.54 ± 0.36 for IBC versus −5.38 ± 0.35 for DCIS) and lower in the *n* = 8 G-to-A PIK3CA-mutated IBC samples when compared with G-to-A PIK3CA-mutated DCIS samples (Mean ± SEM: −6.52 ± 1.66 for IBC versus −6.14 ± 0.74 for DCIS). Although the majority of samples with a PIK3CA mutation were ER+, there was no significant interaction effect between ER status and the absence or presence of the two types of tested PIK3CA mutations (*p* = 0.46 for DCIS and *p* = 0.20 for IBC). 

## 3. Discussion

APOBEC3B has been identified as an important factor in the evolution of breast cancer [8]. In a recently published pan-tissue, pan-cancer analysis of RNA-seq data specific to the seven APOBEC3 genes in 8951 tumors, 786 cancer cell lines and 6119 normal tissues, *APOBEC3B* consistently demonstrated its association with proliferative cells and processes, in contrast to other APOBEC3s, especially *APOBEC3G* and *APOBEC3H*, which were revealed as more immune cell related [9]. Our current data showed that *APOBEC3B* mRNA is already upregulated in the in situ stage of breast cancer, which is in line with the high genomic resemblances between DCIS and IBC [18]. In a study we performed earlier, we observed higher mRNA levels of *APOBEC3B* in breast cancer metastasis as compared to the corresponding primary tumor [14], supporting our hypothesis that, already starting from DCIS, breast cancer progression is associated with deregulated expression of *APOBEC3B*. 

Tumors with upregulated *APOBEC3B* demonstrate a higher mutational load, which could explain the aggressive behavior of these tumors [7,12]. Two hotspot G-to-A mutations in exon 9 of the—often mutated in breast cancer—PIK3CA gene (E542K and E545K) are thought to be generated by *APOBEC3B* induced C-to-T (G-to-A) transitions [11]. Whether *APOBEC3B* is still needed once the mutations are present needs further investigation. In the study of Kosumi et al., *APOBEC3B* expression in esophageal squamous cell carcinoma was significantly correlated with PIK3CA mutations in exon 9 [10]. However, no correlation was found between *APOBEC3B* expression and PIK3CA mutations status in a Japanese breast cancer cohort [15]. Although PIK3CA mutations are known to be more prevalent in ER+ cases [19], and thus might have been a confounder in our analysis, we found no significant difference in the distribution of wild-type and mutated PIK3CA in ER+ and ER− cases. In our cohort, *APOBEC3B* levels were decreased in specifically the G-to-A PIK3CA-mutated IBC samples when compared with wild-type PIK3CA IBC tumors. In the synchronous DCIS counterpart, however, there was no difference in *APOBEC3B* levels between mutated and PIK3CA wild-type tumors. This might suggest that, in contrast to DCIS, the invasive tumors no longer need *APOBEC3B* to proliferate and metastasize. Previous studies reported elevated *APOBEC3B* mRNA levels in breast cancers with otherwise aggressive characteristics, including high histological grade and lack of estrogen expression [7,13,15]. This is consistent with our subgroup analysis, which showed higher *APOBEC3B* levels in synchronous DCIS and IBC of ER− tumors as compared to ER+ tumors. However, in our study, no significant correlations were found between *APOBEC3B* levels and histological grade and/or tumor diameter. This could be due to the fact that the majority of our samples were high grade.

This is the first study evaluating *APOBEC3B* levels within DCIS and co-existing IBC, including different breast cancer subtypes. However, our study has several limitations, such as the relatively small size of our cohort and the analysis of a limited mRNA panel only, with the main focus on *APOBEC3B*. Since upregulated *APOBEC3B* is associated with higher mutational load, evaluation of the mutational status of additional markers besides PIK3CA will be interesting. Another limitation is that *APOBEC3B* is also expressed by inflammatory cells, which could have influenced our data because we performed manual microdissection, and thus contamination with inflammatory cells was not completely avoidable. Although IBC expressed slightly higher levels of *PTPRC* (the gene for leukocyte antigen CD45) than DCIS, there was no correlation between *APOBEC3B* and *PTPRC* mRNA levels. Based on this analysis, it seems unlikely that the number of inflammatory cells biased our data. 

Increased insight in molecular mechanisms that contribute to DCIS progression will improve the development of a personalized treatment strategy for patients with DCIS. APOBEC3B could be a potential therapeutic target since it is non-essential, but it has an active enzymatic activity that may be inhibited [7]. Patients with DCIS could therefore benefit from such therapeutic molecules by inhibiting tumor evolution. Concept inhibitors have already been developed for the related enzyme APOBEC3G [20,21]. Additional clinical and pharmaceutical assays are necessary to develop and explore the potential benefit of APOBEC3B inhibitors. 

## 4. Materials and Methods

### 4.1. Patient Materials

Fifty-three patients with synchronous DCIS and IBC were enrolled. We used coded leftover patient material in accordance with the Code of Conduct of the Federation of Medical Scientific Societies in the Netherlands (http://www.federa.org/codes-conduct). This article is approved by the Medical Ethics Committee of the Erasmus MC (approval number MEC 02.953). According to national guidelines, no informed consent was needed for this study. 

Formalin-fixed-paraffin-embedded (FFPE) hematoxylin and eosin (H&E)-stained whole sections of excision specimens were collected and reviewed by two pathologists (Carolien H. M. van Deurzen and Shusma C. Doebar). Histopathological features included the grade of IBC [22], IBC diameter, ER, PR and Her2 status, and grade of DCIS [23]. Tumors were divided into subtypes based on immunohistochemistry (ER, PR and Her2), including the following 5 categories: ER+/PR high/Her2−; ER+/PR− or low/Her2−; ER+/any PR/Her2+; ER−/PR−/Her2+; ER−/PR−/Her2−. ER was considered positive when at least 10% of the tumor cells were positive, irrespective of intensity, according to national guidelines (https://richtlijnendatabase.nl). Low PR was defined as ≤ 20% [24]. Immunohistochemical HER2 expression was scored according to international guidelines [25]. Equivocal cases were evaluated by silver in situ hybridization.

### 4.2. RT-qPCR

RNA was extracted from tissue areas composed of at least 50% IBC or DCIS cells and analyzed by RT-qPCR as described before [14,17]. In brief, these cells were obtained by microdissection from FFPE tissue, which was performed with a sterile needle under a stereomicroscope. RNA was extracted from these cells using the Qiagen (Hamburg, Germany) AllPrep DNA/RNA FFPE Kit according the manufacturer’s instructions. RNA concentrations were measured with a Nanodrop 2000 system. cDNA was generated from 50 ng/µL cDNA and was generated for 30 min at 48 °C with the RevertAid H minus kit (Thermo Fisher Scientific, Breda, The Netherlands) and gene-specific pre-amplified with Taqman PreAmp Master mix (Thermo Fisher Scientific) for 15 cycles, followed by Taqman probe based real-time PCR according to the manufacturer’s instructions in a MX3000P Real-Time PCR System (Agilent, Amsterdam, The Netherlands). The following intron-spanning gene expression assays (all from Thermo Fisher Scientific) were evaluated: *APOBEC3B*, assay ID: hs00358981_m1; *EPCAM*, hs00158980_m1, and *PTPRC*, hs00236304_m1. Messenger RNA levels were quantified relative to the average expression of 2 reference genes (*GUSB*, hs9999908_m1 and *HMBS*, hs00609297_m1) using the delta Cq (average Cq reference genes −Cq target gene) method. According GeNorm and NormFinder, the average of these two reference genes was the most stable expressed across our samples (M-value = 0.59, SD = 0.29). Also, when taking the different groups into account, the inter and intra variation was the lowest for the average of our 2 reference genes (SD = 0.19 for the NormFinder analysis across the control, DCIS and IBC groups and SD = 0.24 for the NormFinder analysis across the ER/PR/Her2 groups). Samples with an average reference gene expression of Cq > 25 were considered to be of insufficient RNA quality and excluded from further analysis, together with their paired samples. A serially diluted RNA pool of FFPE breast tumor samples was included in each experiment to evaluate the linear amplification and efficiencies for all genes included in the panel and absence of amplification in the absence of reverse transcriptase. All gene transcripts were equally efficient amplified (range 94–106%) and were negative in the absence of reverse transcriptase. A summary of the performance of our assays on these serially diluted samples is shown in Appendix A. 

### 4.3. PIK3CA Mutation Status

PIK3CA mutation status and VAFs were measured as described before [17]. In brief, DNA was extracted from the same micro-dissected FFPE tissues used for RNA extraction using the Qiagen (Hamburg, Germany) AllPrep DNA/RNA FFPE Kit. The SNaPshot Multiplex System for SNP Genotyping (Thermo Fisher Scientific) was used to identify samples positive for PIK3CA hotspot mutations in exon 9 and exon 20. Next, we used digital PCR (dPCR) to validate the SNaPshot results and quantify the relative number of PIK3CA-mutated copies (of E542K, E545K in exon 9 and H1047R and H1047L in exon 20) in both the DCIS and IBC component of those patients with a PIK3CA mutation identified by SNaPshot analysis.

### 4.4. Statistical Analyses

GeNorm and NormFinder [26,27], present in GenEx qPCR data analysis software (version 6.1, MultiD, Götenborg, Sweden), were used to assess the stability of our reference genes. SPSS version 24 was used for the statistical analyses. Because our APOBEC3B mRNA data were not normally distributed (skewness −1.01 ± 0.33 and −1.75 ± 0.33, kurtosis 0.80 ± 0.64 and 3.60 ± 0.64 for DCIS and IBC, respectively), we only used non-parametric tests. The Wilcoxon Signed Ranks Test was used to compare levels in paired DCIS and IBC and unpaired analyses were performed using the Wilcoxon or Mann-Whitney *U* Test or the Fisher Exact Probability Test for contingency tabled data. Continuous variables were analyzed by the Spearman Rank Correlation test. *p*-values ≤ 0.05 were considered statistically significant.

## 5. Conclusions

In conclusion, our results indicate that *APOBEC3B* mRNA is similarly upregulated in DCIS and IBC, but declines in PIK3CA-mutated IBC, which suggests that APOBEC3B plays a role in the early stages of breast carcinogenesis. Since APOBEC3B is a gain-of-function mutagenic enzyme, it could be a candidate for therapeutic targeting in an early, non-invasive stage of breast cancer.

## Figures and Tables

**Figure 1 cancers-11-01062-f001:**
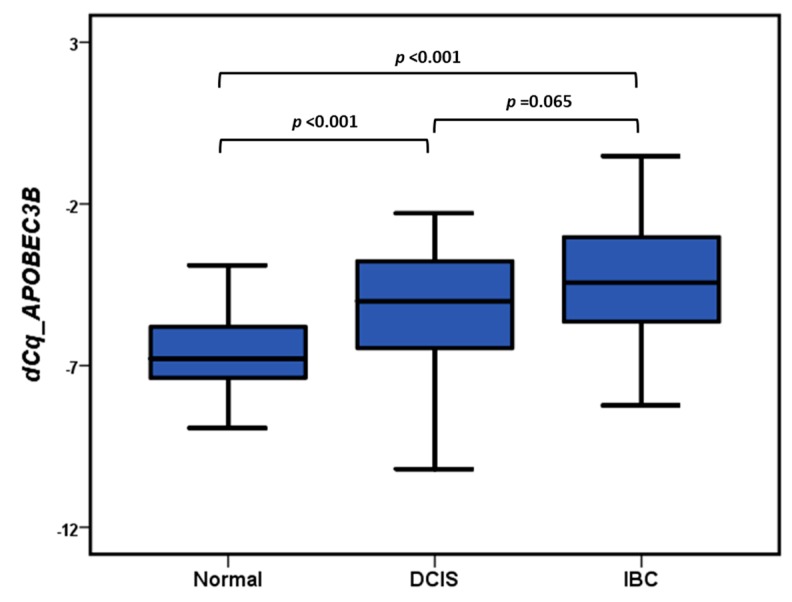
Boxplots of APOBEC3B mRNA expression levels in paired normal, DCIS and IBC (*n* = 53). Differences between normal, DCIS and IBC were analyzed by the Mann-Whitney *U* test.

**Figure 2 cancers-11-01062-f002:**
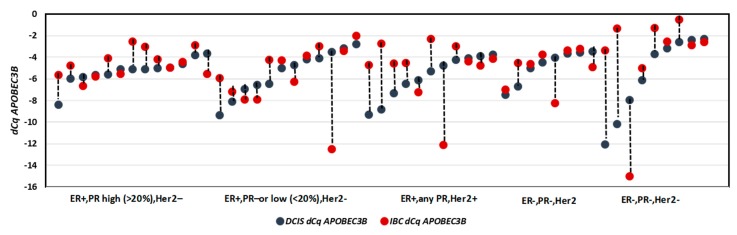
APOBEC3B expression levels in paired DCIS and IBC (*n* = 53). Wilcoxon Signed Ranks Test *p* = 0.082.

**Figure 3 cancers-11-01062-f003:**
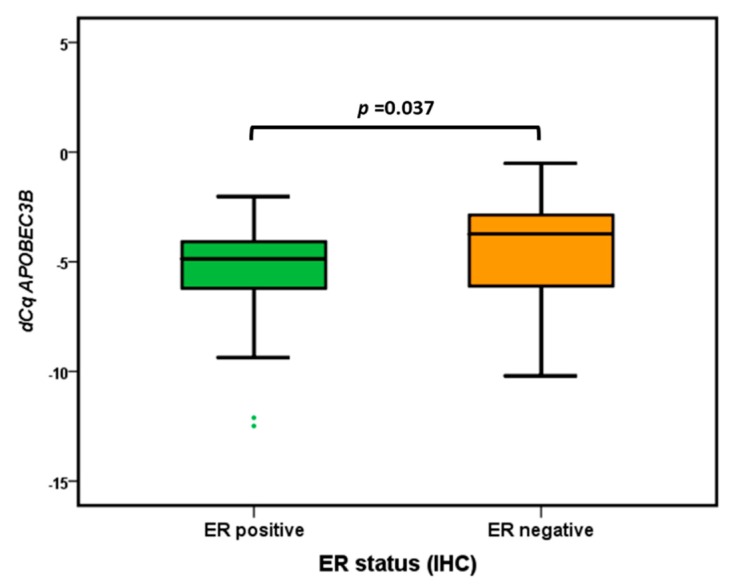
Boxplots of Apolipoprotein B mRNA Editing Enzyme, Catalytic Polypeptide-Like 3B (APOBEC3B) mRNA expression levels according to ER status. The difference between ER+ and ER− cases was analyzed by the Mann-Whitney *U* test.

**Figure 4 cancers-11-01062-f004:**
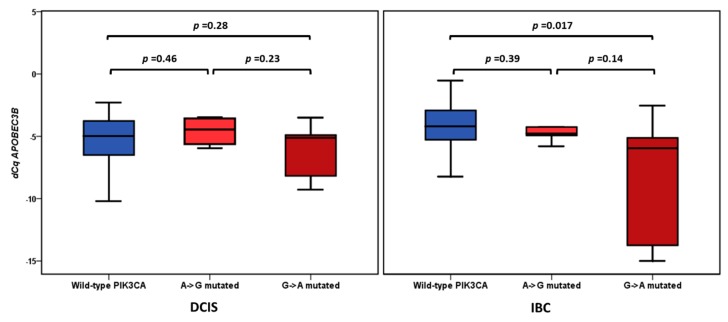
Boxplots of APOBEC3B mRNA expression levels according PIK3CA mutation status. The differences between wild-type (blue boxes) and mutated (red boxes) PIK3CA cases were analyzed by the Mann-Whitney *U* test.

**Table 1 cancers-11-01062-t001:** Clinicopathological features of patients with ductal carcinoma in situ (DCIS) and adjacent invasive breast cancer (IBC) (*n* = 53).

Characteristic	*n*	(%)
**Age at diagnosis**	53	
years, median (range)	(28–102)	
**Type of surgery**		
Breast-conserving surgery	24	45.3
Mastectomy	29	54.7
**Grade DCIS**		
1	1	49.1
2	19	39.6
3	33	7.5
**Grade IBC**		
1	1	49.1
2	21	39.6
3	31	7.5
**Tumor size**		
≤2 cm	28	49.1
>2–5 cm	21	39.6
>5 cm	4	7.5
Missing	0	3.8
**Subtypes based on immunohistochemistry**		
ER+/PR high/Her2−	13	24.5
ER+/PR− or low/Her2−	12	22.6
ER+/any PR/Her2+	11	20.8
ER−/PR−/Her2+	8	15.1
ER−/PR−/Her2−	9	17.0

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
