# Peer review of "APOBEC3B Gene Expression in Ductal Carcinoma In Situ and Synchronous Invasive Breast Cancer"

_cancers, 2019, doi:10.3390/cancers11081062_

Round 1

Reviewer 1 Report

The authors describe an evaluation of APOBEC3B expression in DCIS and IBC. This is of import, since APOBEC3B activity has been associated with mutations in several cancer types, including breast cancer. The work is straightforward, sound and I don't have any suggestions for improving this manuscript

Author Response

The authors describe an evaluation of APOBEC3B expression in DCIS and IBC. This is of import, since APOBEC3B activity has been associated with mutations in several cancer types, including breast cancer. The work is straightforward, sound and I don't have any suggestions for improving this manuscript

Our response:

We thank the reviewer for appreciating our study.

Reviewer 2 Report

By showing a significant expression of APOBEC3B in the early non-invasive stage of breast cancer, the authors in this study wish in the future that patients can benefit from therapeutic targeting of APOBEC3B in the DCIS stage just before the IBC stage. 

However, in order to make the conclusion of this study more concrete, the authors should answer the following question.

Please, explain the reason why the VAF value of PIK3CA oncogenic mutations (induced by APOBEC3B) in DCIS is higher than that in IBC with a more progressed stage, even though there is no difference in APOBEC3B level between IBC and DCIS.   

In the line 77, use “between”, instead of “between and”.

Author Response

By showing a significant expression of APOBEC3B in the early non-invasive stage of breast cancer, the authors in this study wish in the future that patients can benefit from therapeutic targeting of APOBEC3B in the DCIS stage just before the IBC stage.

However, in order to make the conclusion of this study more concrete, the authors should answer the following question.

Please, explain the reason why the VAF value of PIK3CA oncogenic mutations (induced by APOBEC3B) in DCIS is higher than that in IBC with a more progressed stage, even though there is no difference in APOBEC3B level between IBC and DCIS.

Our response:

The reviewer is right that overall we observed no statistically significant difference in APOBEC3B mRNA levels between IBC and DCIS (see also Figures 1 and 2). However, our subgroup analysis showed that APOBEC3B levels were decreased in specifically the G-to-A PIK3CA mutated IBC samples when compared with wild-type PIK3CA IBC tumors. In the synchronous DCIS counterpart, however, there was no difference in APOBEC3B levels between mutated and PIK3CA wild-type tumors (see also Figure 4).

In addition, the correlation of PIK3CA VAF with APOBEC3B showed a negative Spearman Rank correlation in IBC (rs = -0.33, p = 0.001, n = 53). For the DCIS cases there was no such correlation (rs = 0.02, p = 0.89, n = 53). So, in summary, we did observe a difference in APOBEC3B expression between IBC and DCIS when we analysed our data in relation to PIK3CA mutational status.

However, to better explain, we added the following in the result section of our revised manuscript on page 4, lines 114-118:

Albeit not statistically significant, APOBEC3B mRNA levels were higher in the n= 39 PIK3CA wild-type IBC samples when compared with the PIK3CA wild-type DCIS samples (Mean ± SEM: -4.54 ± 0.36 for IBC versus -5.38 ± 0.35 for DCIS) and lower in the n=8 G-to-A PIK3CA mutated IBC samples when compared with G-to-A PIK3CA mutated DCIS samples (Mean ± SEM: -6.52 ± 1.66 for IBC versus -6.14 ± 0.74 for DCIS).

In the line 77, use “between”, instead of “between and”.

Thank you for noticing this mistake, we have now changed this in our revised manuscript.

Reviewer 3 Report

Summary of Paper:

The current paper investigates APOBEC3B  as the underlying mechanism of cellular transformation of ductal carcinoma in situ (DCIS) to invasive breast cancer (IBC). Gene expression expression analysis of APOBEC3B in synchronous DCIS and IBC and surrounding normal was conducted from 53 patients, with tumors categorized on ER, PR, Her2 and PIK3CA status. There no differences in APOBEC3B between wild-type versus mutated PIK3CA DCIS, APOBEC3B was higher in wild-type versus PIK3CA-mutated IBC. Depending on the overlap of the primer sets used for qPCR experiments, the gene expression data suggest APOBEC3B may be upregulated in DCIS and driving cellular transformation to IBC.

Comments:

There was no difference in APOBEC3B mRNA expression between DCIS and IBC (unpaired 68 Mann-Whitney U Test p = 0.065” (lines 68, 69) should be “There was no statistically significant difference in APOBEC3B mRNA expression between DCIS and IBC (unpaired Mann-Whitney U Test p = 0.065”)

The figures graph display raw data instead of ranked data and the ranked data were used for the statistical analysis because the raw data were transformed to ranks. Please show the median calculated on the ranked data, which can mislead the reader to believe that graphed data are the same values the statistical tests were calculated on. Showing the raw data is fine but the raw data were converted to a scale to permit statistical tests and those statistical tests needs to be indicated on the ranked data instead of the raw data, using median (not average). For accurate statistical analysis, please use a nonparametric an omnibus test, such as Kruskal-Wallis H test by ranks to compare all three groups (control, DCIS, and IBC) and if the Kruskal-Wallis H is significant then subsequently calculate Mann-Whitney U or Wilcoxon sign ranked tests compare between two groups

In the methods state clearly and report the kurtosis and skewness of the raw data are not normally distributed or skewed and if any transformation were calculated or if nonparametric tests were chosen instead to analyze the gene expression data.

Is there a correlation between levels of APOBEC3B detected by qRT-PCR and PIK3CA gene mutations in individual patients across cancer stages? The authors need to calculate and report it. This would be a better indication of association.

What is the idea behind comparing APOBEC3B levels in ER+ vs ER- patients? The authors didn’t explain well.

What are the expression levels of the other APOBEC3 family members? Evaluation of Thermo’s Hs00358981_m1 indicates these primers will amplify APOBEC3A, and APOBEC3A/B hybrid mRNAs. In many cancers APOBEC3A/B chimeric transcripts are quite common (references).  Also, on a related technical limitation, since the specimens are FFPE samples, which have RNA decay in 5 prime end, the authors will have more consistent results using primers designed for the 3’-end region of the mRNA.

The current data raise and support the possibility that Gene expression experiments cannot make this determination. Expression changes supported the possibility and hypothesis that APOBEC3B is a putative driver for transformation at DCIS stage but the current gene expression  data doesn’t show APOBEC3B is neither necessary nor sufficient for tumorigenesis.

APOBEC3 Atlas – reference e in Nucleic Acids Research 47:1178-1194, 2019 – show expression patterns of it across numerous cancers,. Please references the seminal paper pan-cancer gene expression analysis in >8,951 tumors, including breast cancer, 786 cancer cell lines, and 6119 normal samples

Please include the dilution curves for Taqman probe analysis as supplemental data and please acknowledge the caveats choice of using as a reference probes. The Warburg Effect of cancer preferring carbohydrate metabolism as a its primary energy substrate is a potential confound of using GAPDH, because it is a major metabolic step in the carbohydrate pathway (glycolysis, step 6). Therefore, it cannot be used as an universal “housekeeping gene” as a stable, consistent reference probe for cancer studies. Please show that carbohydrate metabolism is stable across the control, DCIS, & IBC groups.

For transparent replication of the experiment and the results please provide the panel of control genes without these critical details the experiments and data analysis cannot be replicated by other laboratories.

Round 2

Reviewer 3 Report

The authors addressed all my concerns well except a minor one. I now accept the paper.